# Integrating Object Detection Modality into Visual Language Model for Enhanced Autonomous Driving Agent

**Linfeng He**[1*]**, Yiming Sun**[2*]**, Sihao Wu**[1]**, Jiaxu Liu**[1†]**, Xiaowei Huang**[1]
[1]Department of Computer Science, University of Liverpool, United Kingdom
[2]School of Computer Science, University of Nottingham, United Kingdom
{sihao.wu, jiaxu.liu, xiaowei.huang}@liverpool.ac.uk
{hermannhelf, sym990908}@gmail.com

## Abstract

In this paper, we propose a novel framework for enhancing visual comprehension in autonomous driving systems by integrating visual language models (VLMs) with additional visual perception module specialised in object detection. We extend the Llama-Adapter architecture by incorporating a YOLOS-based detection network alongside the CLIP perception network, addressing limitations in object detection and localisation. Our approach introduces camera ID-separators to improve multi-view processing, crucial for comprehensive environmental awareness. Experiments on the DriveLM visual question answering challenge demonstrate significant improvements over baseline models, with enhanced performance in ChatGPT scores, BLEU scores, and CIDEr metrics, indicating closeness of model answer to ground truth. Our method represents a promising step towards more capable and interpretable autonomous driving systems. Possible safety enhancement enabled by detection modality is also discussed.

## 1 Introduction

The rapid advancements in autonomous driving systems have led to an increased focus on developing end-to-end models capable of handling complex driving scenarios. Despite significant progress, current approaches still face challenges in generalisation, especially when faced with rare or unseen situations. Moreover, the ability to interact with human users and provide explanations for the model's decisions is crucial for building trust and acceptance of autonomous vehicles. To address these challenges, the recently introduced DriveLM challenge Sima et al. (2023) aims to leverage the power of vision-language models (VLMs) and large language models (LLMs) in the context of autonomous driving. By combining the visual understanding capabilities of VLMs with the reasoning and natural language processing abilities of LLMs, DriveLM seeks to improve generalisation and enable interactive communication between autonomous vehicles and human users. Inspired by the DriveLM framework Sima et al. (2023), this paper presents a novel approach that integrates additional modalities into the LLMs to enhance its perception and reasoning capabilities for autonomous driving tasks.

Our method builds upon the Llama-Adapter Zhang et al. (2023a), a parameter-efficient fine-tuning approach that allows for the incorporation of task-specific knowledge into the pre-trained LLM. The common practice for image perception capability in Llama-Adapter is to incorporate a pre-trained image embedder, specifically the CLIP Radford et al. (2021a) model with trainable vision transformers (ViT) Dosovitskiy et al. (2020) to generate adaptation queries. The queries are then

---

[*]Equal Contribution, [†]Project Lead

projected and appended onto layer-wise token embeddings. However, such a perceptual network has critical limitation, which is based on the employment of CLIP. Although CLIP is effective at capturing global contextual information, it struggles to accurately detect and locate objects in the image, as it is primarily trained on perceptual prompts rather than position-level annotations. To address this limitation, we propose the integration of a detection network into the Llama-Adapter framework. The detection network leverages pretrained YOLOS Fang et al. (2021b) and postfix vision transformers to process multi-view camera inputs and generate rich feature representations that accurately capture object-specific information, such as positions and bounding boxes. Moreover, we introduced trainable ID-separator token to address confusion of object-camera relationship to concatenated YOLOS output.

By incorporating the detection network, our approach enables: 1) The capability to sense local, fine-grained details in the driving scene, complementing the global understanding provided by the perceptual network. 2) Understanding of different viewpoints of BEV images via trainable ID tokens and detection-result-oriented fine-tuning. 3) enhancement of scene understanding robustness and potential defence against vulnerabilities from visual modality.

We believe that a standalone detector is crucial for autonomous driving, which helps to ensure safer and more reliable decision-making by improving the capability to manage complex scenarios such as the detection of pedestrians, vehicles, and traffic signs. To evaluate the effectiveness of our approach, we conducted extensive experiments on the DriveLM challenge, comparing our model's performance against state-of-the-art baselines. We demonstrate significant improvements in the ChatGPT score, the BLEU score, and the CIDEr score (see Sec. 4.2).

The main contributions of this paper can be summarised as follows: *(1)* We identify the limitations of relying solely on CLIP-based features for perception in the Llama-Adapter framework and propose the integration of a detection network to overcome these challenges. *(2)* We leverage pretrained YOLOS and vision transformers in the detection network to accurately capture object-specific information and enhance the perceptual capabilities of the Llama-Adapter. *(3)* We demonstrate significant improvements in multiple matrices on overall driving performance through extensive experiments on the DriveLM challenge, showcasing the benefits of integrating the detection network.

## 2 Related work

### 2.1 LLM-based autonomous driving

Fu et al. (2024) and Wen et al. (2023) explore the potential of leveraging large language models (LLMs) to imbue autonomous driving systems with human-like reasoning, interpretation, and memorization capabilities. They argue that such knowledge-driven approaches can address the limitations of traditional optimization-based and modular systems when dealing with complex scenarios and long-tail corner cases. The DiLu framework proposed by Wen et al. Wen et al. (2023) combines Reasoning and Reflection modules to enable decision-making based on common-sense knowledge and continuous evolution, demonstrating strong performance and generalization in experiments. DriveGPT4 Xu et al. (2023) presents an interpretable end-to-end autonomous driving system that utilizes large language models. They develop a new visual instruction tuning dataset for interpretable autonomous driving with the assistance of ChatGPT, and mix-finetune DriveGPT4 on this dataset. Mao et al. Mao et al. (2023) introduce GPT-Driver, which reformulates motion planning as a language modeling problem and utilizes the GPT-3.5 model as a motion planner. Their prompting-reasoning-finetuning strategy exhibits superior planning performance, generalization, and interpretability compared to existing methods on the nuScenes dataset. These works highlight the potential of LLMs in enabling more human-like reasoning and decision-making in autonomous driving systems, addressing the challenges faced by traditional approaches.

### 2.2 VLM-based autonomous driving

Recent works explore leveraging visual information to enhance autonomous driving systems. Sima et al. (2023) introduces Graph VQA, a task modeling graph-structured reasoning in autonomous driving, and propose DriveLM-Data and DriveLM-Agent, a VLM-based baseline. Their experiments demonstrate the potential of integrating VLMs into driving systems to enhance generalization and interactivity. Chen et al. (2023) proposes a novel object-level multimodal LLM architecture that fuses

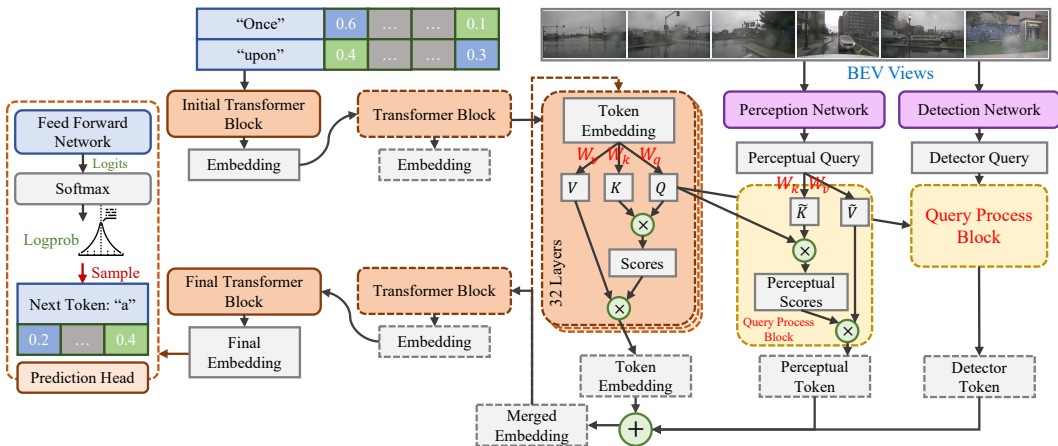

Figure 1: Schematic of our VLM Autonomous Driving framework. Projection Network and Detection Network process BEV view camera images respectively for a QA pair. Their outputted token are then merged into hidden state of each layer in decoder layers in the language model to pass to next layer.

vectorized numeric modalities with a pre-trained LLM to improve context understanding in driving situations. They also present a new dataset and evaluation metrics for driving question answering. DriveVLM Tian et al. (2024) introduces an autonomous driving system that integrates a chain-of-thought process with vision-language models for enhanced scene understanding and planning. They further propose DriveVLM-Dual, a hybrid system combining DriveVLM with traditional perception and planning modules. Wayve (2023) propose LINGO, a visual analytics interface that helps identify and reduce bias in natural language task instructions for pre-trained language models. Their user study demonstrates that LINGO promotes the creation of more challenging and linguistically diverse tasks with lower instruction bias.

In contrast to these works that focus on integrating VLMs into driving pipelines, our paper aims to move beyond purely perceptual tasks by proposing a framework that integrates modalities more than visual and language to enable more comprehensive reasoning, interpretation, and memorization in an autonomous driving agent.

## 3 Method

### 3.1 Model Architecture

The model architecture is based on traditional decoder block language model architecture. Let the model input $\mathbf{x} \in \mathbb{R}^{N \times d_{\text{seq}}}$, where $N$ is the length of current sequence and $d_{\text{seq}}$ denote the initial token dim. Let the embedding dim be $d_{\text{emb}}$, the decoder-only transformer without token prediction head is formulated by a function $f_{\text{trans}} : \mathbb{R}^{N \times d_{\text{seq}}} \to \mathbb{R}^{N \times d_{\text{emb}}}$. Assume we have an $L$-layer transformer, within each layer, the decoder block $f_{\text{block}} : \mathbb{R}^{N \times d_{\text{emb}}} \to \mathbb{R}^{N \times d_{\text{emb}}}$ is defined by

$$f_{\text{block}}^{(l)}(\mathbf{z}) := W_o(\text{normalize}(W_q(\mathbf{z})W_k(\mathbf{z})^\top)W_v(\mathbf{z})) \in \mathbb{R}^{N \times d_{\text{emb}}}, \tag{1}$$

where $W_q, W_k, W_v, W_o : \mathbb{R}^{N \times d_{\text{emd}}} \to \mathbb{R}^{N \times d_{\text{emd}}}$ are linear transformations within attention blocks. With an initial linear transformation $f_{\text{init}} : \mathbb{R}^{N \times d_{\text{seq}}} \to \mathbb{R}^{N \times d_{\text{emb}}}$, the transformer $f_{\text{trans}}$ is defined by

$$f_{\text{trans}}(\mathbf{x}) := f_{\text{block}}^{(L)}(\cdots f_{\text{block}}^{(2)}(f_{\text{block}}^{(1)}(f_{\text{init}}(\mathbf{x})))) \in \mathbb{R}^{N \times d_{\text{emb}}}, \tag{2}$$

and the final embedding is obtained via

$$\mathbf{z}_{\text{final}} = f_{\text{trans}}(\mathbf{x}). \tag{3}$$

The next token is predicted by feeding $\mathbf{z}_{N-1,:}^{(L)} \in \mathbb{R}^{1 \times d_{\text{emb}}}$ into prediction head.

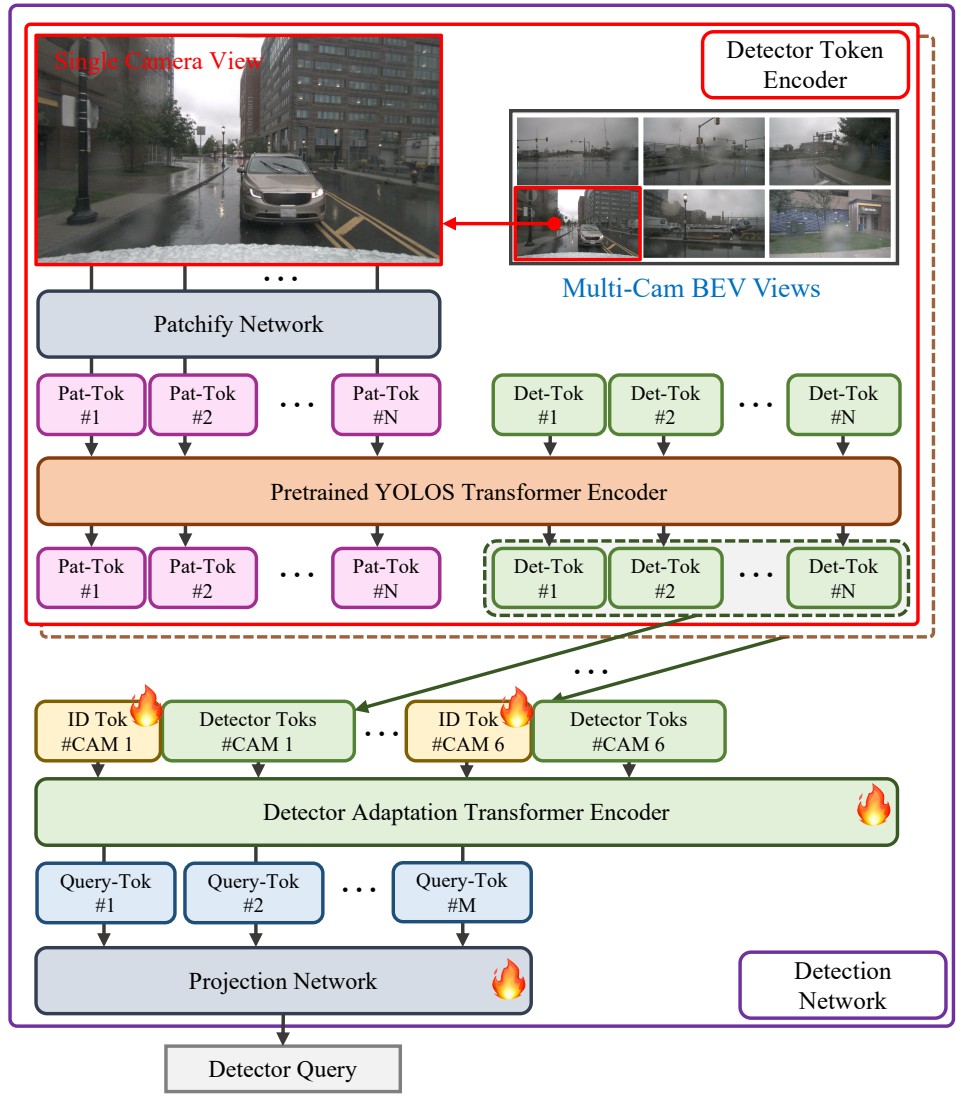

Figure 2: Detection network for detector query generation. Each image is processed into tokens separately then concatenated together with trainable ID-separator tokens. $f_{\text{proj}}$ is implemented with the Detector Adaptation Transformer Encoder $\mathbb{R}^{M \times d_{\text{yolos}}} \to \mathbb{R}^{M \times d_{\text{yolos}}}$ and the Projection Network $\mathbb{R}^{M \times d_{\text{yolos}}} \to \mathbb{R}^{M \times d_{\text{emb}}}$

## 3.2 Integration of Textual and Visual Data

To incorporate visual data, our model interfaces the Transformer Blocks with a Perception Network and a Detection Network. The Perception Network processes raw images to generate visual embedding, while the Detection Network focuses on identifying specific features within the images. The integration process can be formalised as:

$$\mathbf{Q}_{\text{percept}} \in \mathbb{R}^{M \times d_{\text{emb}}}, \mathbf{Q}_{\text{detect}} \in \mathbb{R}^{M \times d_{\text{emb}}} = f_{\text{percept}}(\mathbf{X}_{\text{imgs}}), f_{\text{detect}}(\mathbf{X}_{\text{imgs}}), \qquad (4)$$

where $f_{\text{percept}}(\mathbf{X}_{\text{imgs}}), f_{\text{detect}}(\mathbf{X}_{\text{imgs}})$ are respectively the Perception and Detection Network (as in Fig. 2). $\mathbf{Q}_{\text{percept}}$ and $\mathbf{Q}_{\text{detect}}$ are respectively the Perceptual and Detector Query as shown in Fig. 1. with the queries, we transform them according to the following procedure

$$\text{Tok}_\star = W_o(\text{normalize}(W_q(\mathbf{z})W_k(\mathbf{Q}_\star)^\top)W_v(\mathbf{Q}_\star)) \in \mathbb{R}^{N \times d_{\text{emb}}}. \qquad (5)$$

Replacing $\star$ in Eq. (5) by $\{\text{percept}, \text{detect}\}$ we obtain the $\text{Tok}_{\text{percept}} \in \mathbb{R}^{N \times d_{\text{emb}}}$ and $\text{Tok}_{\text{detect}} \in \mathbb{R}^{N \times d_{\text{emb}}}$, which have exactly the same shape as textual tokens $f_{\text{block}}^{(l)}(\mathbf{z})$ in Eq. (1). These tokens

are then added to the textual tokens processed by each Transformer Block (Eq. (1)). We fine-tune the parameters within Eq. (5) on downstream context to facilitate this integration, which allows the model to augment the textual information with visual context, leading to enriched representations.

### 3.3 Detection Network with Camera-Specific Adaptation

A distinctive aspect of our model is the Detection Network's adaptation to multiple camera inputs. Each camera input is associated with a unique set of ID-separator Tokens to distinguish between different sources. Same as in Fig. 2, the process on obtaining Detector Query $\mathbf{Q}_{\text{detect}}$ is described as

$$\mathbf{Q}_{\text{detect}} = f_{\text{proj}}(\|_{i=1}^{6} \left[ \text{Tok}_{\text{ID}}^{\text{cam}_i} \in \mathbb{R}^{1 \times d_{\text{yolos}}} \| \text{Tok}_{\text{yolos}}^{\text{cam}_i} \in \mathbb{R}^{N \times d_{\text{yolos}}} \right]) \in \mathbb{R}^{M \times d_{\text{emb}}}, \qquad (6)$$

where $\|$ denotes concatenation, $\text{Tok}_{\text{ID}}^{\text{cam}_i}$ denotes ID-seperator Token (a trainable tensor) corresponding to the $i$th camera, $\text{Tok}_{\text{yolos}}^{\text{cam}_i}$ denotes YOLOS (Fang et al. (2021b))' output token given input image of the $i$th camera. Here, we took the output of the last layer of YOLOS' encoder blocks, and $d_{\text{yolos}}$ denotes dimension of this output. $N$ is the length of the outputted tokens, which is fixed and corresponds to the number of detection tokens configured in YOLOS [2] Feeding $\mathbf{Q}_{\text{detect}}$ to Eq. (5) gives the adapted token $\text{Tok}_{\text{detect}}$. Finally $\text{Tok}_{\text{detect}}$ is passed through a projection layer, which is a transformation $f_{\text{proj}} : \mathbb{R}^{d_{\text{emb}}} \to \mathbb{R}^{d_{\text{emb}}}$. $M$ in this case is equally $6 + 6N$. In such as way, the Detector Query $\mathbf{Q}_{\text{detect}}$, enriched with camera-specific visual information, is integrated into the Transformer's processing pipeline.

Our use of trainable ID-separator tokens to separate groups of tokens in concatenation could be seen as an extension to trainable adapter prompt technique used in Zhang et al. (2023a). In Zhang et al. (2023a), llama-adapter used trainable adapter prompt to store guidance to language model learnt through fine-tuning. We used a similar structure differently to encode the token group information, prefixed each token sequence with unique "trainable prompt" which we call ID-separator, separating object from different images.

### 3.4 Next Word Prediction from Merged Token Embeddings

According to our framework Fig. 1, we merge layer-wise Perception/Detection tokens with contextual tokens as

$$\text{Tok}_{\text{merged}}^{(l)} = f_{\text{block}}^{(l)}(\mathbf{z}^{(l)}) + \mathbf{g}_{\text{percept}} \text{Tok}_{\text{percept}}^{(l)} + \mathbf{g}_{\text{detect}} \text{Tok}_{\text{detect}}^{(l)}, \qquad (7)$$

where $\text{Tok}_{\star}^{(l)}$s are generated via Eq. (5). $\mathbf{g}_{\star}$s are trainable zero gates as introduced in Zhang et al. (2023a). The output $\text{Tok}_{\text{merged}}^{(l)} \in \mathbb{R}^{N \times d_{\text{emb}}}$ from the final Transformer Block is passed to the Prediction Head, which generates the next word in the sequence based on both the textual and visual contexts. The predicted next word

$$\text{NextWord} = \text{Sample}(\text{Softmax}(\text{FFN}((\text{Tok}_{\text{merged}}^{(L)})_{N-1,:}))), \qquad (8)$$

where $\text{FFN} : \mathbb{R}^{d_{\text{emb}}} \to \mathbb{R}^{d_{\text{vocab}}}$ here is a feed forward network transforming the vector in model dimension $\mathbb{R}^{d_{\text{emb}}}$ to the vocabulary size $\mathbb{R}^{d_{\text{vocab}}}$ (typically 32000). The softmax is required after FFN for inspecting the sampling-probability of each word within the vocabulary dictionary, and finally a sampler is applied for generating the next word. The framework repeat this procedure for continued word generation until the [eos] token is generated.

## 4 Experiments and discussions

### 4.1 Experiment details

**Dataset.** We used NuScenes(Caesar et al. (2019)) for finetuning and experiment. The dataset is structured into training set, test and validation set. The training set consist of 2896 QA pairs, each corresponds to a scene with 6 camera images. We tested on test dataset of length 66 and 2987, as well as on our validation set of length 15480.

---

[2]In YOLOS, each detection token corresponds to one possible detected object, embedding information of possible classes and bounding box.

Table 1: Test result for sample datasets, which include 66 Q&A pairs.

| Experiment | ACCURACY | CHATGPT | MATCH | BLEU_1 | BLEU_2 | BLEU_3 | BLEU_4 | ROUGE_L | CIDEr | FINAL_SCORE |
|---|---|---|---|---|---|---|---|---|---|---|
| Ground Truth | 1.0 | 100 | 100.00 | 0.999 | 0.0010 | 0.000100 | 0.000032 | 1.00 | 1.92 | 0.90 |
| Ground Truth (only Tag 0 correct) | 1.0 | 79.44 | 27.5 | 0.058 | 0.0002 | 0.000038 | 0.000015 | 0.09 | 0.12 | 0.58 |
| DriveLM-Agent | 0.0 | 65.11 | 28.25 | 0.049 | 0.0002 | 0.000036 | 0.000014 | 0.08 | 0.09 | 0.32 |
| Our Method (Llama-Adapter) | 0.0 | 80.44 | 30.25 | 0.041 | 0.0002 | 0.000034 | 0.000014 | 0.07 | 0.09 | 0.38 |
| Our Method (Yolos) | 0.0 | **76.11** | **32.50** | **0.059** | **0.0002** | **0.000039** | **0.000015** | 0.09 | **0.12** | **0.38** |

**Implementation Details.** We used llama-adapter-v2-multimodal-7b (Zhang et al. (2023b)) and YOLOS Fang et al. (2021a) for our model implementation. In fine-tuning, we used a learning rate of $1 \times 10^{-5}$, with weight decay set to 0.05 and batch size of 2. The fine-tuning is carried out in 2 steps – we first fine-tuned the detection token projection network $f_{\mathrm{proj}}$ and ID-separator tokens in whole and then second bias in llama. We did not train or fine-tune other parts of the architecture(visual network comes pre-trained in llama-adapter-v2-multimodal-7b).

### 4.1.1 Experiments

We list the experiments as follows

**Ground Truth** Using ground truth in place of model output; used as reference of best scores a model can get.

**Ground Truth (only Tag 0 correct)** Using ground truth in place of model output only for questions in NuScenes dataset with tag 0 [3]

**DriveLM-Agent** Using DriveLM-Agent (Sima et al. (2023)) to answer the questions, and compare output with ground truth.

**Our Method (Llama-Adapter)** Using llama-adapter-v2-multimodal-7b (Zhang et al. (2023b)) to answer questions and compare with ground truth.

**Our Method (Yolos)** Using our architecture integrating Yolos-based detection network to answer the questions, and compare output with ground truth.

### 4.1.2 Metrics

The metrics employed in the experiments are as follows

**Accuracy** 1 if the output string is exactly the same as the ground truth answer, 0 otherwise. Average is taken on all (output, ground truth) pairs.

**ChatGPT** Score given by ChatGPT, with prompt "rate the following answer based on the correct answer", providing it ground truth and output.

**Match** percentage of points in groundtruth (e.g., "[1.,2.]") that are "close" to any point in question (with $L^1$ distance less than 16)

**BLEU_{1,2,3,4}** BLEU score for text similarity evaluation with {1,2,3,4} n-gram precisions.

**ROGUE_L** The ROGUE-L(Lin and Och (2004)) longest-common-sequence-based score for text similarity evaluation.

**CIDEr** The CIDEr (Vedantam et al. (2014)) score for image description evaluation.

**Final_Score** The final score is a weighted sum of the previous scores, defined as:

$$
\text{Final\_Score} = 0.4 \times \frac{\text{ChatGPT}}{100} + 0.2 \times \frac{\text{match}}{100} + 0.2 \times \text{accuracy}
$$
$$
+ 0.2 \times \left( \frac{\sum_{i=1}^{4} \frac{\text{BLEU}_i}{4} + \text{ROGUE\_L} + \frac{\text{CIDEr}}{10}}{3} \right). \tag{9}
$$

---

[3]This means this questions is about perception, in the 5 categories [perception, prediction, planning, behavior, motion]

Table 2: Test result for self-evaluation datasets, which include 2987 Q&A pairs.

| Experiment | ACCURACY | CHATGPT | MATCH | BLEU_1 | BLEU_2 | BLEU_3 | BLEU_4 | ROUGE_L | CIDER | FINAL_SCORE |
|---|---|---|---|---|---|---|---|---|---|---|
| DriveLM-Agent | - | - | - | - | - | - | - | - | - | - |
| Our Method (Llama-Adapter) | 0.0 | 65.55 | 18.59 | 0.041 | 0.0002 | 0.000034 | 0.000014 | 0.076 | 0.082 | 0.3057 |
| Our Method (Yolos) | **0.2966** | 58.243 | **21.1484** | **0.1078** | **0.0333** | **0.0105** | **0.199** | 0.0093 | **0.2632** | **0.3548** |

Table 3: Test result for final validation datasets, which include 15480 Q&A pairs.

| Experiment | ACCURACY | CHATGPT | MATCH | BLEU_1 | BLEU_2 | BLEU_3 | BLEU_4 | ROUGE_L | CIDER | FINAL_SCORE |
|---|---|---|---|---|---|---|---|---|---|---|
| DriveLM-Agent | 0.0 | 67.75 | 18.83 | 0.238 | 0.0995 | 0.0367 | 0.011200 | 0.19 | 0.0074 | 0.3284 |
| Our Method (Llama-Adapter) | 0.0 | 57.84 | 15.56 | 0.247 | 0.1046 | 0.0114 | 0.000014 | 0.20 | 0.0086 | 0.2825 |
| Our Method (Yolos) | 0.0 | **57.95** | **21.19** | 0.18 | 0.0609 | 0.0212 | 0.006684 | **0.20** | 0.0047 | 0.2919 |

## 4.2 Analysis of Experiment Results

From Tab. 1-3, we conclude the followings

**Good Performance with Smaller Sized Test Dataset.** The main difference caused by size of test dataset is distribution of question types and variety of scenario/objects. As test datasets are not used in training, it's likely that over-fitting is not the cause; rather, this indicates that the addition of object detection modality could be enhancing answering of different types of questions differently. A next step in this direction is to evaluate the model's performance on questions with different tags seperately, as well as visualizing characteristics of question it got wrong/right. Apart from shift of question category distribution, larger dataset could also be challenging when more types of objects are getting involved, whereas YOLOS can only label 92 classes of objects.

**Generally Better Than Llama-Adapter.** In all experiments our model performs better than its predecessor (Llama-Adapter with only CLIP visual network), indicating that object detection modality, if not generally useful in visual question answering, enhance more answers than those it corrupts.

**High Match Score.** Our model's Match score in all experiments are particularly high as compared to other metrics, as well as other models. Match score specifically measures how accurate points in answers are compared to ground truth, such as current or future coordinates of objects. High Match score indicates that addition of object detection modality enhanced precise object position detection and/or subsequent prediction/reasoning.

**Low CIDEr Score on Larger Dataset.** Our model have especially lower CIDEr score on larger datasets than llama-adapter, while higher CIDEr score on smaller datasets. Since CIDEr score measures accuracy of image captioning, it indicates that addition of YOLOS object detection modality corrupts the model's image captioning ability on larger datasets. Our guess is while CLIP is pre-trained on a very large WebImageText dataset (Radford et al. (2021b)), YOLOS is pre-trained on COCO (Lin et al. (2015)), which is much smaller and have only 92 classes of objects. Therefore, it's lacking captioning capability could be corrupting CLIP's work, such as by classifying detected object wrong or way too simply(e.g., car instead of black Nissan), while the test dataset gets larger and more types of objects start appearing in the dataset.

## 4.3 Using additional modalities to defend against backdoor attack

As shown by Perez et al. (2022); Liu et al. (2024); Ni et al. (2024), LLM/VLM could be subject to various visual attacks, exploiting vulnerabilities from the visual modality, such as modality red-teaming, backdoor attack with trigger objects, adding to risks in autonomous driving setting.

With detection modality, and perhaps more modalities to add, we can put up defense against such attacks by *(1)* Reaping more robust understanding with the driving scene backed and double checked with multiple modalities and *(2)* Distributing visual modality's potentially pivoting influence to model's output over to detection modalities and other potential modalities.

With object detection modality as example:

**Higher-precision Scene Understanding** In Ni et al. (2024), one example of such backdoor attack is by showing a photo of a red balloon, to trigger the model to output sudden high acceleration. The attack is done with poisonous training example that associates red balloon with sudden acceleration. With addition of object detection modality, the scene will no longer be percieved as only "scene with

red balloon", but "scene with red balloon on $< 20.0, 40.0 >$". While this probably won't eliminate such attack, as defense to backdoor attack probably would always involve consensus reasoning, the addition of position feature (which is effective as seen in our Match score) probably can confuse the attack pattern, making the attack effective in less cases(such as "only red balloon on $< 20.0, 40.0 >$ triggers acceleration").

**Distributing Influences over Other Modalities** As seen in the dip of CIDEr score with addition of object detection modality, although CLIP have advanced image description capability, its influence can be watered down with other modalities. An attack targeting a specific modality can thereby be wakened, as the target modality's responsibility will be shared with other untargeted and perhaps unaffected modalities. For example, if a modality using only lidar point cloud information is integrated, it would not be affected by an attack only poisoning visual encoder.

## 5   Conclusion

This paper introduces a novel framework enhancing visual comprehension in autonomous driving systems by integrating large language models. We combine a YOLOS-based detection network with a CLIP perception network in the Llama-Adapter framework, improving object recognition and scene understanding. Our approach, featuring innovative camera ID-separators for multi-view processing, shows significant performance gains on the DriveLM challenge. This work advances the development of more capable and interpretable autonomous driving systems that effectively merge language models with visual perception. Our experimental results show competitive performance across various metrics, including ChatGPT scores and BLEU scores. While there's room for improvement in accuracy, our approach shows promise in combining LLMs with specialised visual modules for autonomous driving. The addition of detection modalities can also potentially enhance driving agent's safety. Future work should focus on further integrating detection and perception networks, enhancing multi-view processing, explore the enhancement of safety against various attacks and expanding the dataset for more comprehensive evaluations. This research represents a step towards more capable and interpretable autonomous driving systems that leverage the strengths of both language models and visual perception modules.

## 6   Limitations

**Dependency on High-Quality Dataset.** Experiment is carried out only on a portion of Nuscenes dataset.The current evaluation is confined to specific datasets and scenarios. The model's performance in diverse, real-world environments remains untested. The model's performance heavily relies on the quality and variety of the data used for training. Any biases or deficiencies in the dataset could adversely affect the system's reliability and decision-making.

**Model Complexity and Computation.** The framework have good scalability in term of finetuning, inherited from llama-adapter. The large base models(Llama, CLIP, YOLOS) are frozen, only adapter networks are trained. Inference-wise, increase of computation cost is linear to number of adapted networks - a fixed number more step will be computed in decoder layers, and a fixed number more steps will be computed in processing. This might limit the deployment of such systems in real-world scenarios where computational resources are constrained.

**Scalability Issues.** The scalability of the proposed approach to larger, more diverse datasets and across different geographic regions has not been fully explored. This includes challenges related to adapting the system to various driving conditions and legal requirements. Autonomous driving systems must adhere to evolving regulations and standards.

**Interpretability and Error Analysis.** While the model incorporates LLMs for better interpretability, the complex interactions between textual and visual data within the model make it challenging to pinpoint the source of errors and understand decision-making processes in depth.

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
