# OpenReview forum: "Integrating Object Detection Modality into Visual Language Model for Enhanced Autonomous Driving Agent"
_NeurIPS.cc/2024/Workshop/SafeGenAi — SafeGenAi Poster_

### Official Review · Reviewer_GFWT · 2024-10-09
**The paper's novelty lies around adding a YOLOS-based detection network, which helps detect and locate objects (like cars, pedestrians, etc.) more accurately compared to previous methods.**

**Rating:** 7
**Confidence:** 5

**Review:**

Pros:
- The paper's novelty lies around adding a YOLOS-based detection network, which helps detect and locate objects (like cars, pedestrians, etc.) more accurately compared to previous methods.
-  It introduces a new technique (trainable ID-separators) that helps the system handle inputs from multiple cameras, which is crucial for getting a comprehensive view of the environment while driving.
- the paper addresses an important downside of the vanilla adapter, and by combining the detection network with the Llama-Adapter architecture, the model enhances its ability to understand fine details in driving scenes, making decision-making safer and more reliable.

Cons:
- The experiments are based on a specific dataset, and it's not clear how it generalizes. The paper doesn’t fully explore how well the model scales to larger datasets or more complex environments, which could be important for real-world applications of autonomous driving.
- the paper uses very specific models (YOLO, llama) - however it's not clear how it scales to state of the art VLMs (https://arxiv.org/abs/2405.17247)

---

### Official Review · Reviewer_YBt3 · 2024-10-09
**Novel work - Multicamera specific additions to Explainable Vision Language Model.**

**Rating:** 8
**Confidence:** 2

**Review:**

The research work presents additions to existing Vision Language Models for Autonomous driving application.

Quality:
1. The research paper introduces multicamera processing with Camera ID, and CLIP model for object detection, on exising VLM for Autonomous Driving.
2. Presents through evaluation metrics like ChatGPT scores, BLEU scores, and CIDEr metrics.

Clarity:
- Pros:
1. The work shows two new methods, one using CLIP and the other using Yolos, and showing that Yolos performs better due to the object detection aspect of the architecure.
- Cons:
1. The Computation requirement for inference such as VRAM, inference time per frame, is not mentioned, which is critial information for realtime Autonomous Driving application.
2. A sample text prompt input related to the Autnomous driving scenario and output of the YOLOS detections could have been mentioned in the figures or in the text.
3. The qualitative evaluation of the difference in output text with and without the addition of the camera ID separators could have been presented.


Significance:
- Interpretability: The CLIP and Llama additions of this pipline shows explainablity of the model's predictions, which is key for gaining trust in autonomous driving technologies.
- Safety: By improving the ablilty to detect and localize pedestrians, vehicle, road signs and other artifacts on the road, the system can reduce accidents caused by misrepresentation of the scene objects.
- Con: As mentioned in the Limitations section, the system maybe computationally heavy to be depolyed directly on the Autonomous Vehicle's realtime perception stack.

---

### Official Review · Reviewer_hAAR · 2024-10-09
**VLM-Based Driving with Object Detection - Good idea with room for improvement and relevance**

**Rating:** 3
**Confidence:** 3

**Review:**

Review Overview:\
This paper provides a method for VLM-based driving integrating object detection. Although the motivation is sound, I believe the results are slightly weak, can improve in presentation, and the paper is not very relevant to the SafeGenAI workshop.

This review follows the sections of the paper. Strengths (+) and weaknesses (-) are noted for each section.

Introduction:\
\+ The introduction to the use of VLM in driving is good.
\- Are the authors certain that it's CLIP-based features that cause perception issues as per the first contribution claim? Visualizing how the language attends to the detections or performing ablations with and without the CLIP-based features would help strengthen this claim.\
\- Consider rephrasing the third claim. The meaning of "improvements in multiple matrices" is not clear.

Related Work:\
\- Line 81 could use more specificity: "challenges faced by traditional approaches." What challenges are being referred to here?\
\- Line 26 and line 98 indicate that additional modalities were used, but the work has not specified yet which additional modalities (lidar, radar, IMU?). That should be stated clearly. Upon finishing the paper, I'm still unsure what modalities are referring to. Are you referring to detection as a modality? I would suggest a more precise term such as "Output task" or "intermediate task."

Method:\
\- Figure 1: Is $f_\text{Adaptation}$ not shown in the figure 1? Presumably it goes before detection network, $f_\text{Detection}$?\
\- After reading lines 126-130 a few times, it is still unclear what ID-separator is. Consider rephrasing this explanation. "trainable prompt" meaning it's a label for a given object? "separating object from different images" - can't there be multiple objects within the same image? Is only one image given one ID-separator? "Specific group an object belongs to" meaning like the class of an object? Maybe walk through an example or provide a figure with an example of a learned ID-separator. Is the ID-separator the same as the ID-tokens of equation (4)? To my understanding, the ID-tokens were for different camera viewpoints, not to separate objects from different images.\
\- What loss functions are being used? How do you train the detection network?

Experiments:\
\- Tables 1, 2, and 3 are very difficult to interpret. What is Ground Truth and only Tag 0? Are the scores under ChatGPT the accuracies reported compared to ChatGPT? What's the accuracy column? How is the final score determined? The paper could benefit from a few sentences describing/comparing the metrics. I don't think ROUGE_L is mentioned anywhere.\
\- Are detection bounding boxes or scores ever outputted? Is it possible the method is never detecting objects at all?

Safety:\
\- This section is quite weak and seems like an attempt to make this paper relevant to SafeGenAI workshop. This section could benefit from analysis of how specifically the proposed architecture either improves safety or suffers from safety concerns. Also, measuring results regarding tests of safety, or visualizations of when this method behaves in an unsafe manner would help.

Conclusion/Limitations:\
\+ The conclusion provides a good summary, and the limitations are well written and acknowledged.